# Evidence of the Berezinskii-Kosterlitz-Thouless phase in a frustrated magnet

Ze Hu[1,13], Zhen Ma[2,3,13], Yuan-Da Liao[4,5,13], Han Li[6,13], Chunsheng Ma[1], Yi Cui[1], Yanyan Shangguan[2], Zhentao Huang[2], Yang Qi [7,8,9 ✉], Wei Li [6,10 ✉], Zi Yang Meng [4,11,12 ✉], Jinsheng Wen [2,9 ✉] & Weiqiang Yu[1 ✉]

The Berezinskii-Kosterlitz-Thouless (BKT) mechanism, building upon proliferation of topological defects in 2D systems, is the first example of phase transition beyond the Landau-Ginzburg paradigm of symmetry breaking. Such a topological phase transition has long been sought yet undiscovered directly in magnetic materials. Here, we pin down two transitions that bound a BKT phase in an ideal 2D frustrated magnet $TmMgGaO_4$, via nuclear magnetic resonance under in-plane magnetic fields, which do not disturb the low-energy electronic states and allow BKT fluctuations to be detected sensitively. Moreover, by applying out-of-plane fields, we find a critical scaling behavior of the magnetic susceptibility expected for the BKT transition. The experimental findings can be explained by quantum Monte Carlo simulations applied on an accurate triangular-lattice Ising model of the compound which hosts a BKT phase. These results provide a concrete example for the BKT phase and offer an ideal platform for future investigations on the BKT physics in magnetic materials.

[1] Department of Physics and Beijing Key Laboratory of Opto-electronic Functional Materials and Micro-nano Devices, Renmin University of China, Beijing 100872, China. [2] National Laboratory of Solid State Microstructures and Department of Physics, Nanjing University, Nanjing 210093, China. [3] Institute for Advanced Materials, Hubei Normal University, Huangshi 435002, China. [4] Beijing National Laboratory for Condensed Matter Physics and Institute of Physics, Chinese Academy of Sciences, Beijing 100190, China. [5] School of Physical Sciences, University of Chinese Academy of Sciences, Beijing 100190, China. [6] School of Physics, Key Laboratory of Micro-Nano Measurement-Manipulation and Physics (Ministry of Education), Beihang University, Beijing 100191, China. [7] Center for Field Theory and Particle Physics, Department of Physics, Fudan University, Shanghai 200433, China. [8] State Key Laboratory of Surface Physics, Fudan University, Shanghai 200433, China. [9] Collaborative Innovation Center of Advanced Microstructures, Nanjing University, Nanjing 210093, China. [10] International Research Institute of Multidisciplinary Science, Beihang University, Beijing 100191, China. [11] Department of Physics and HKU-UCAS Joint Institute of Theoretical and Computational Physics, The University of Hong Kong, Pokfulam Road, Hong Kong, SAR, China. [12] Songshan Lake Materials Laboratory, Dongguan, Guangdong 523808, China. [13]These authors contributed equally: Ze Hu, Zhen Ma, Yuan-Da Liao, Han Li. ✉email: qiyang@fudan.edu.cn; w.li@buaa.edu.cn; zymeng@hku.hk; jwen@nju.edu.cn; wqyu_phy@ruc.edu.cn

Topology plays an increasingly important role in understanding different phases and phase transitions in correlated quantum matters and materials. One prominent example is the Berezinskii–Kosterlitz–Thouless (BKT) mechanism in two-dimensional (2D) systems[1–5], which is associated with the binding and unbinding of topological defects. The BKT transition cannot be characterized by conventional order parameters and constitutes the earliest example of phase transition beyond the Landau–Ginzburg paradigm of spontaneous symmetry breaking. Historically, the BKT mechanism was introduced in the *XY* spin model and long predicted to occur in magnetic thin films[1,4]. In experiments, signatures of the BKT transition have been observed in superfluid helium films[6], as well as in 2D superconducting films[7,8] and arrays[9]. Regarding the original proposal in layered *XY*-type magnets, despite intensive efforts[10–15], direct and unambiguous observation of the BKT transition is still lacking. One major obstacle is the three-dimensional (3D) couplings in the magnets, although weak, will inevitably enhance the confining potential of vortices[15], leading to 3D ordering that masks the BKT transition. Therefore, it is of fundamental interest to find and identify BKT materials that could overcome the obstacle and study the topology-related low-energy dynamics.

Recently, a layered frustrated rare-earth antiferromagnet TmMgGaO$_4$[16–18] was reported to ideally realize the triangular-lattice quantum Ising (TLI) model[19]. The relatively large interlayer distance of 8.3774 Å along the *c* axis gives rise to excellent two dimensionality[17] and no sign of conventional 3D phase transition was observed in either specific heat or magnetic susceptibility measurements. Nevertheless, it was reported from neutron scattering that TmMgGaO$_4$ ordered below ~1 K into an antiferromagnetic phase with a sixfold symmetry breaking[16,18], which closely resembles the ground state of the TLI model originated from an order-by-disorder mechanism[20,21]. At higher temperatures, the effective *XY* degrees of freedom emerge and the BKT mechanism is expected to come into play[21].

In TmMgGaO$_4$, each Tm$^{3+}$, with a $4f^{12}$ electron configuration and a spin–orbit moment $J = 6$, forms a non-Kramers doublet due to the crystal-electric-field splitting. The doublet is well separated from the rest levels by about 400 K[16] and can thus be regarded as an effective spin-1/2. There is further a fine energy splitting within the doublet, induced by the local trigonal crystal field[17], acting as an intrinsic "transverse field" applied on the effective spin. From the magnetization measurements[16–18], one observes that Tm$^{3+}$ ions contribute highly anisotropic Ising-type moments with $J_z = \pm 6$ along the *c* axis, resulting in an effective out-of-plane *g*-factor ~ 13.2[16,19]. On the contrary, the effective in-plane *g*-factor and dipolar moment in the *ab* plane are negligible.

Facilitated by this feature in TmMgGaO$_4$, in this work we employed nuclear magnetic resonance (NMR), a sensitive low-energy probe, to detect the BKT phase. We applied a moderate in-plane field of 3 T, which is adequate to collect the $^{69}$Ga NMR signals and, at the same time, hardly disturbs the low-energy electronic states of the material. This is important, as in the TLI model that is believed to accurately model TmMgGaO$_4$[19], the BKT phase can be fragile against out-of-plane fields[22–24], thus posing a challenge to NMR measurements. Taking advantage of the fact that in-plane moment in TmMgGaO$_4$ is mostly multipolar[16,19], our NMR experiments with in-plane fields, which merely couple to the nuclear spins, can clearly identify the BKT phase in the material.

As shown in Fig. 1, from our NMR measurements of the spin-lattice relaxation rate $1/T_1$, we identify $T_U \simeq 1.9$ K and $T_L \simeq 0.9$ K, which represent the upper- and lower-BKT transitions, where a critical BKT phase resides at zero magnetic field in between the high-*T* paramagnetic and low-*T* antiferromagnetic phases. This finding is further substantiated by our scaling analysis of the

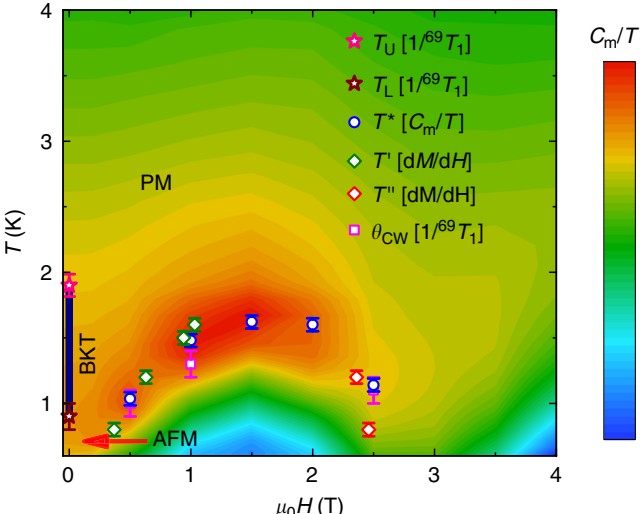

**Fig. 1 Phase diagram of TmMgGaO$_4$ under out-of-plane magnetic fields.** Under zero field, there are paramagnetic (PM), BKT, and antiferromagnetic (AFM) phases. The $T_U$ ($T_L$) is the upper (lower) BKT transition temperature, determined from the plateau structure in the NMR spin-lattice relaxation rate $1/^{69}T_1$ (see Fig. 2c for details). The BKT phase between $T_U$ and $T_L$ is illustrated by the solid vertical line, while the AFM regime is indicated by the arrow. The contour background depicts the magnetic specific heat $C_m/T$ at various fields and temperatures, with data adapted from ref. [16] and plotted in logarithmic scale. $T^*$ corresponds to the maximum of $C_m/T$ at each field, signifying the position of strong magnetic fluctuations. $T'$ ($T''$) denotes the temperature at a specific field where a peak is found in the differential susceptibility $dM/dH$, shown in Fig. 3b. The Curie–Weiss temperature $\theta_{CW}$ is obtained from the $1/^{69}T_1T$ (see Supplementary Fig. 1). Remarkably, $T^*$, $T'$, $T''$, and $\theta_{CW}$ all collapse to the same phase boundary between the BKT-like regime and AFM phase. A magnetically ordered phase is supposed to lie below the dome-like boundary. Errors represent 1 SD throughout the paper.

measured susceptibility data near $T_L$, as well as the simulated NMR and susceptibility data using large-scale quantum Monte Carlo (QMC) calculations.

## Results

**NMR probe of the BKT phase.** The obtained NMR spectra with an in-plane magnetic field $\mu_0H = 3$ T are shown in Fig. 2a at representative temperatures. To better resolve the magnetic transition, the hyperfine shifts $^{69}K_n$ of the spectra were analyzed and plotted in Fig. 2b as a function of temperature. Upon cooling, $^{69}K_n$ peaks at about 0.8–0.95 K and then starts to drop at lower temperatures. Therefore, the ordering temperature is determined to be $T_L \simeq 0.9$ K, consistent with neutron scattering experiments[16,18]. In addition, both the second moments (width of the NMR spectra) and the third moments (asymmetry of the spectra) of the spectra change dramatically below ~ 2 K, suggesting the onset of local hyperfine fields enhanced by the static or quasi-static magnetic ordering (Supplementary Fig. 4). These two characteristic temperatures signal the two-step melting of magnetic order through two BKT transitions, suggesting an intermediate floating BKT phase in the system. We suspect that there is some inhomogeneity of the local hyperfine fields, which is very likely caused by the quenched disorder from Mg/Ga site mixing[16], although no significant influence on the electronic and more importantly the magnetic properties is seen (see more detailed discussions in Supplementary Note 4).

The spin-lattice relaxation rate $1/T_1$ provides a highly sensitive detection of low-energy spin fluctuations[25–29], and thus the BKT

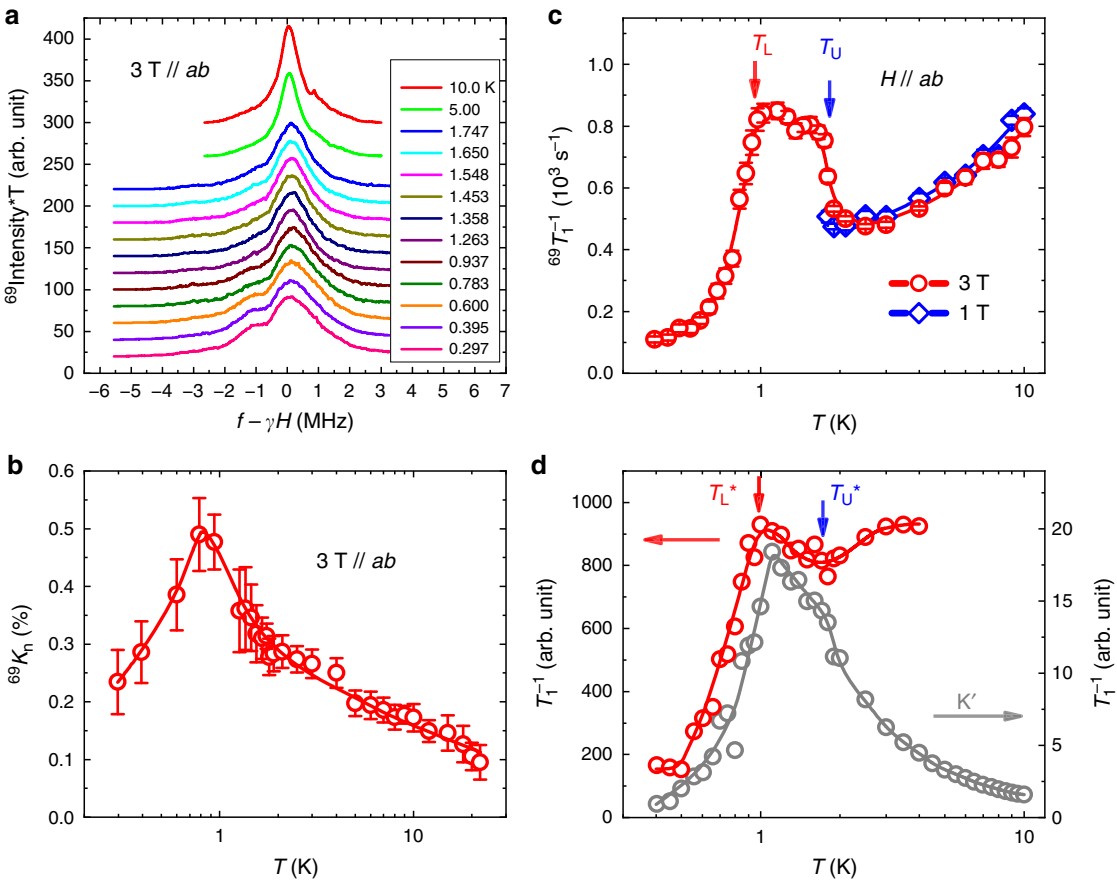

**Fig. 2 NMR spectra and spin-lattice relaxation rates of TmMgGaO$_4$. a** $^{69}$Ga NMR spectra at different temperatures under a 3 T in-plane field, with zero frequency corresponding to $\gamma H = 30.692$ MHz. Data are shifted vertically for clarity. At high temperatures, the spectra are roughly symmetric, whereas for $T \leq 2$ K, a shoulder-like structure can be resolved on the left side of the main peak. **b** NMR hyperfine shift $^{69}K_n = (\bar{f}/\gamma H - 1) \times 100\%$ as a function of temperature, where $\bar{f}$ is the average frequency of each spectrum. **c** NMR spin-lattice relaxation rate $1/^{69}T_1$ vs. temperature measured under in-plane fields of 3 T and 1 T. A plateau-like feature, characterizing strong magnetic fluctuations, is observed between $T_L \simeq 0.9$ K (lower-BKT transition) and $T_U \simeq 1.9$ K (upper BKT transition). **d** $1/T_1$ data computed from the dynamical spin–spin correlation function with contributions from all momentum points [c.f., Eq. (3) and see the hyperfine form factor in Supplementary Note 2] in the Brillouin zone (left scale) and from $K'$ point in the vicinity of the $K$ point (right scale), through large-scale QMC simulations (see "Methods").

transition. In Fig. 2c, we show the $1/^{69}T_1$ obtained under in-plane fields $\mu_0 H = 3$ T and 1 T, which reflects intrinsic spin fluctuations with zero out-of-plane field. At 3 T, $1/^{69}T_1$ first decreases upon cooling from 10 K then suddenly increases below $T_U \approx 1.9$ K, indicating the onset of strong low-energy spin fluctuations. The data at 1 T show similar behaviors. Below $T_L \simeq 0.9$ K, $1/^{69}T_1$ drops sharply, consistent with the onset of the magnetic ordering as also inferred from the hyperfine shift. Here, $1/^{69}T_1$ is dominated by the gapped long wavelength excitations in the ordered state. At the magnetic phase transition, a peaked feature in $1/T_1$ develops, caused by the gapless low-energy spin fluctuations with diverging correlation length. Remarkably, at temperatures between $T_U \simeq 1.9$ K and $T_L \simeq 0.9$ K, $1/^{69}T_1$ exhibits a plateau-like structure, indicating the emergence of a highly fluctuating phase with diverging spin correlations yet no true long-range order, which is the hallmark of a BKT phase[1–5]. Therefore, it is for the first time that such a phase is unambiguously observed in a magnetic crystalline material.

Our unbiased QMC simulations on the TLI model of the material (see "Methods"), with accurate coupling parameters determined in ref. [19], quantitatively justifies the existence of the BKT phase between $T_L$ and $T_U$. We computed $1/T_1$ and compared with the experiment below. Figure 2d shows the calculated $1/T_1$ by including fluctuations from all momentum

points in the Brillouin zone (cf. Supplementary Fig. 2) and compare to that from $K'$ (around the $K$ point at the corner of Brillouin zone). The former shows a decrease upon cooling below 4 K and then an upturn above $T_U^* \simeq 2$ K, followed by a rapid decrease below $T_L^* \simeq 1$ K. These behaviors are in excellent agreement with the measured $1/^{69}T_1$. The latter reflects gapless excitations of the $XY$ degrees of freedom emergent in the BKT phase, where the calculated $1/T_1$ from $K'$ exhibits an anomalous increase down to $T_U^*$, below which the increment slows down. The contribution to $1/T_1$ near the $K$ point reaches a maximum at $T_L^*$ and drops rapidly below it. The absence of critical spin fluctuations at momentum away from the K point suggests that the measured $1/^{69}T_1$ below 2 K is mainly contributed by excitations around the $K$ point (see Supplementary Note 3).

Overall, the quasi-plateau behaviors in the QMC results and the two characteristic temperature scales are in full consistency with the NMR measurements. This constitutes both strong support for the accurate quantum many-body modeling of the material TmMgGaO$_4$ and also solid proof of the BKT phase therein detected by NMR. Nevertheless, we note that there are still subtle differences between the experimental and numerical data. Needless to say, the real material is always more complicated than our theoretical minimal model. For example, influences from higher crystal-electric-field levels above the non-Kramers

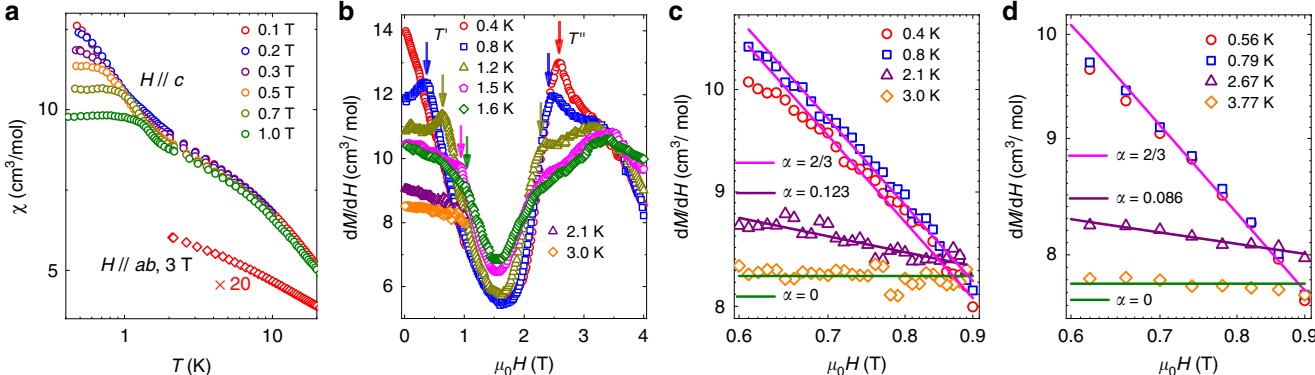

**Fig. 3 Uniform magnetic susceptibility of TmMgGaO₄ and scaling analysis. a** dc susceptibility $\chi$ as functions of temperatures under small out-of-plane ($H//c$) and in-plane ($H//ab$) fields. The latter is multiplied by a factor of 20 for visualizing purpose. The deviation of data below 2 K indicates the entry to the BKT phase and the field-suppression of magnetic correlations. **b** The differential susceptibility $dM/dH$ under out-of-plane fields at different temperatures. The kinks at low fields, as denoted by the down arrows, suggest the transition from the BKT-like phase to the ordered phase (under the dome in Fig. 1) with increasing fields. The peaked features at high fields suggests a quantum phase transition to the polarized phase. **c** Fits of $dM/dH$ to the power-law scaling function $dM/dH \sim H^{-\alpha}$ with $\alpha = 2/3$ for the 0.4 and 0.8 K data, and $\alpha = 0.123$ for 2.1 K data in the log-log scale. The 3 K data follow the $\alpha = 0$ line in the paramagnetic phase. **d** $dM/dH$ by the QMC calculations in the same field and temperature range as in **c**, and fits to the power-law function with exponents $\alpha$, which give consistent results as experiments.

doublet, the interlayer couplings not included in our model calculations, and the lack of knowledge on the precise local hyperfine coupling constant, etc., may explain the difference remaining between Fig. 2c, d.

**Universal magnetic susceptibility scaling.** Magnetic susceptibility $\chi$ measurements were also performed to strengthen the finding of the BKT phase. In Fig. 3a, we show the overall temperature dependence of $\chi$ at different out-of-plane fields. For $T \gtrsim$ 2 K, $\chi$ increases monotonically upon cooling and barely changes with fields. However, for $T \lesssim 2$ K, approximately the upper BKT transition $T_U$ as obtained from the $1/^{69}T_1$ measurements, $\chi$ increases as the field decreases, suggesting the onset of peculiar magnetic correlations. With further cooling, the susceptibility gets flattened with temperature. The magnetization $M(H)$ was further measured at selected temperatures (data shown in Supplementary Fig. 6), and for the sake of clarity, the differential susceptibility $dM/dH$ is plotted in Fig. 3b. At around 2.5 T, a pronounced peak can be observed at low temperature, indicating the existence of a quantum phase transition, and the phase at lower fields should be a magnetically ordered phase, although its precise nature remains to be uncovered. Besides the high-field feature, for temperatures at 0.8 K and above, a kinked feature is clearly resolved on each $dM/dH$ curve at low fields, whereas at 0.4 K, the low-field kink disappears, which posts a question of whether there is a quantum transition or merely a crossover from the zero-field AFM phase to the finite-field ordered phase under the dome in Fig. 1. The temperature and field values indicated by the down arrows in Fig. 3 are denoted as $T'$ and $T''$ in the phase diagram (Fig. 1).

Field-theoretical analysis of the TLI model[23,24] has predicted that upon applying a small out-of-plane field, the differential susceptibility $dM/dH$ shall exhibit a divergent power-law behavior as $dM/dH \sim H^{-\alpha}$ in proximity to the BKT phase. At $T_L$, $\alpha$ has the value of 2/3, which corresponds to a critical exponent $\eta = 1/9$ at the lower-BKT transition and is originated from the sixfold symmetry breaking[23]. The exponent $\alpha$ gradually decreases as temperature increases, and above an intermediate temperature between $T_L$ and $T_U$, $\alpha = 0$ due to non-universal contributions. This is exactly what we observe in Fig. 3c. We fit the $dM/dH$ with the power-law function at different temperatures, with the fitting regime chosen between 0.6–0.9 T. At 0.8 K, $\alpha$ is very close to the expected value of 2/3 (and thus $\eta = 1/9$) at $T_L$, which constitutes a

remarkable fingerprint evidence for the BKT transition. At lower temperatures such as 0.4 K, the exponent is also close to 2/3, because the susceptibility saturates with temperature, as shown in Fig. 3a. At high temperatures, $\alpha$ drops rapidly to a small value 0.12 at 2.1 K and becomes effectively zero at 3.0 K.

Therefore, the susceptibility scaling gives the lower-BKT transition at about 0.8 K and upper transition probably between 2.1 and 3 K, in good agreement with the $T_L$ and $T_U$ estimated from NMR. These results are also fully consistent with our QMC calculations on the susceptibility shown in Fig. 3d. At $T_L$ or lower, $\alpha$ is 2/3, then decreases to a very small exponent 0.086 at 2.67 K, and above 3 K, becomes zero within numerical uncertainty. Such a power-law behavior in $dM/dH$ again signifies the finite-temperature window of the BKT phase with diverging magnetic correlations, which gives rise to the universal power-law scaling of magnetic susceptibility.

## Discussion

We believe the findings in this work are of various fundamental values. Since the original proposal of a BKT phase in magnetic films[3-5], which also triggered the currently thriving research field of topology in quantum materials, tremendous efforts have been devoted to finding the BKT phase in magnetic crystalline materials, yet hindered by the obstacle outlined in the Introduction. Here, benefiting from NMR as a sensitive low-energy probe, and the nearly zero planar gyromagnetic factor in a TLI magnet TmMgGaO₄, we are able to reveal two BKT transitions and a critical BKT phase with an emergent $XY$ symmetry. Together with the power-law behavior of the susceptibility and excellent agreement between our QMC simulation and experiment data, we unambiguously identify the long-sought BKT phase in a magnetic crystalline material.

Many intriguing questions are stimulated, based on the phase diagram in Fig. 1 obtained here. First, what is the nature of the ordered phase under finite fields, are there further exotic phases and transitions in the phase diagram, and will there be a field-induced quantum phase transition at the high-field side of the dome—these are all of great interests to be addressed in future studies. Second, it should be noted that the dynamical properties obtained by QMC calculations in Fig. 2 are computed on a large, while finite-size, 36 × 36 lattice, which already produces $1/T_1$ data in very good agreement with the experimental measurements.

Such a great agreement is surprising, given the possible existence of randomness from Ma/Ga site mixing in the material TmMgGaO$_4$[16], and also the lattice disorder revealed by the large high-temperature second moment of the NMR spectra (Supplementary Note 4). Although the random distribution in intrinsic transverse fields and spin couplings does not seem to alter the low-temperature spin-ordered phase and the sharp spin excitation line shapes[18,19], its intriguing effects on the finite-temperature phase diagram of TLI and also the compound TmMgGaO$_4$ call for further studies.

Third, in the study of BKT transition in superfluid systems, it has been observed experimentally and understood theoretically that additional dissipations also appear above the transition temperature due to fluctuations of vortices[6]. Hence, the plateau of $1/T_1$ we observe may also cover regions slightly above the upper BKT transition temperature. We leave this subtlety to future numerical and experimental efforts. Lastly, in general terms, whether there are other rare-earth magnetic materials in the same family of TmMgGaO$_4$ that, acquire similar 2D competing magnetic interactions from highly anisotropic gyromagnetic factor and unique triangular-lattice structures, and also exhibit the BKT physics, is quite intriguing and calls for future investigations. All these directions are ready to be explored from here.

## Methods

**Crystal growth and susceptibility measurements**. Single crystals were grown by the optical-floating-zone method with an image furnace (IR Image Furnace G2, Quantum Design). The natural cleavage surface of the crystals is the $ab$ plane, which allows us to align the field orientation within an error of 2°. The dc susceptibility was measured in a PPMS VSM (Quantum Design) for temperatures above 2 K and in a He-3 MPMS (Quantum Design) for temperatures ranging from 0.4 to 2 K.

**NMR measurements**. The $^{69}$Ga ($I = 3/2$, $\gamma = 10.219$ MHz/T) NMR spectra were collected with the standard spin-echo sequence, with frequency sweep by a 50 kHz step using a topping tuning circuit. The NMR hyperfine shift was obtained by calculating the change of the first moment of the spectra. The spin-lattice relaxation rate $1/^{69}T_1$ was measured by the inversion-recovery method, where a $\pi/2$ pulse was used as the inversion pulse. The NMR data from 1.8 K and above were measured in a variable temperature insert, and the data from 1.8 K and below were measured in a dilution refrigerator. The weak NMR signal at low fields and the rapid decrease of $^{69}T_2$ upon cooling (Supplementary Note 5) prevented us to measure the $1/^{69}T_1$ for in-plane fields <3 T, with temperature below 1.8 K. Whereas for in-plane fields of 4 T and higher, the sample could not be held in position because of the large anisotropy in the g-factor and unavoidable sample misalignment ($\lesssim$2°). At $T = 1.2$ K, we did not find any change of $1/^{69}T_1$ with two different radio frequency excitation levels (14 mT and 24 mT), and with different frequencies across the NMR line, within the error bar.

**Triangular-lattice Ising model**. At zero field, the intralayer couplings in TmMgGaO$_4$ can be described by the TLI Hamiltonian,

$$H = J_1 \sum_{\langle i,j \rangle} S_i^z S_j^z + J_2 \sum_{\langle\langle i,j \rangle\rangle} S_i^z S_j^z + \sum_i \Delta S_i^x, \quad (1)$$

where $J_1$ and $J_2$ are the superexchange interactions among Tm$^{3+}$, $\langle i,j \rangle$ and $\langle\langle i,j \rangle\rangle$ refer to the nearest neighbors and the next-nearest neighbors, respectively, and $\Delta$ is the energy splitting within the non-Kramers doublet imposed by the crystal field. We have shown that the parameter set $J_1 = 0.99$ meV, $J_2/J_1 \approx 0.05$ and $\Delta/J_1 \approx 0.54$ reproduces the experimental results of the transition temperatures and the inelastic neutron scattering spectra[19].

In the TLI model [Eq. (1)], we can define a complex field $\psi$ as a combination of the Ising (Z) components $m_{A,B,C}^z$ on three sublattices, i.e.,

$$\psi = m_A^z + e^{i2\pi/3} m_B^z + e^{i4\pi/3} m_C^z. \quad (2)$$

Notably, $\psi = |\psi|e^{i\theta}$ is a complex order parameter that represents the emergent $XY$ degree of freedom relevant to the BKT physics in the TLI model.

**QMC calculations**. QMC simulations were performed in the path integral in the $S_{i,\tau}^z$ basis with discretization in space and time. The lattice of $L \times L \times L_\tau$, where $L = 36$ and $L_\tau = \beta/\Delta\tau$ with $\Delta\tau = 0.05$ and $\beta = 1/T$, were simulated with both local and Wolff-cluster update schemes[30,31]. The $1/T_1$ results were obtained by first computing the dynamical spin–spin correlation function $\langle S_i^z(\tau)S_j^z(0) \rangle$ and then acquiring its real-frequency dependence $S(\mathbf{q}, \omega)$ from the stochastic analytic

continuation[32]. We then determined the $1/T_1$ either by summing the contributions close to momentum $K$ or over the entire Brillouin zone, as discussed in the Fig. 2d of the main text,

$$T_1^{-1}(\mathbf{q}) = \frac{1}{L^2} \sum_{\mathbf{q}} |A_{\mathrm{hf}}(\mathbf{q})|^2 S(\mathbf{q}, \omega \to 0), \quad (3)$$

where $A_{\mathrm{hf}}(\mathbf{q})$ is the hyperfine coupling form factor (see Supplementary Note 2). Similar analyses have been successfully applied to the QMC computation of NMR $1/T_1$ for the spin-1/2 and spin-1 chains[33,34].

## Data availability
The data that support the findings of this study are available from the corresponding authors upon reasonable request.

## Code availability
All numerical codes in this paper are available upon request to the corresponding authors.

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

## Acknowledgements

We thank Changle Liu, Rong Yu, Nvsen Ma, and Anders Sandvik for stimulating discussions. We acknowledge the supports from the National Key Projects for Research and Development of China through Grant numbers 2016YFA0300502 and 2016YFA0300504, the National Natural Science Foundation of China through Grant numbers 11574359, 11674370, 11822405, 11674157, 11974036, 11834014, 11874115, and 51872328, RGC of Hong Kong SAR China through Grant number 17303019, Natural Science Foundation of Jiangsu Province with Grant number BK20180006, Fundamental Research Funds for the Central Universities with Grant number 020414380117, and the Research Funds of Renmin University of China. We thank the Center for Quantum Simulation Sciences in the Institute of Physics, Chinese Academy of Sciences, the Computational Initiative at the Faculty of Science and the Information Technology Services at the University of Hong Kong, the Platform for Data-Driven Computational Materials Discovery at the Songshan Lake Materials Laboratory, Guangdong, China, and the Tianhe-I, Tianhe-II, and Tianhe-III prototype platforms at the National Supercomputer Centers in Tianjin and Guangzhou for their technical support and generous allocation of CPU time.

## Author contributions

W.Q.Y. and J.S.W. designed the experiments, with proposals from Y.Q., W.L., and Z.Y.M. Z.M. grew and characterized the single crystals and performed susceptibility measurements and analysis, with help from Y.Y.S.G., Z.T.H., Z.H., and H.L. Z.H., C.S.M., and Y.C. performed NMR measurements and analysis. Y.D.L. and H.L. carried out the large-scale quantum many-body calculations, with the guidance from Y.Q., W.L., and Z.Y.M. W.Q.Y., J.S.W., W.L., Z.Y.M., and Y.Q. wrote the manuscript with comments from all coauthors.

## Competing interests

The authors declare no competing interests.
