## [Peer Review File · Nature Communications]

Reviewers' Comments:

Reviewer #1:

Remarks to the Author:

The authors present a combination of NMR, susceptibility and specific heat data with QMC simulations for the two-dimensional antiferromagnet TmMgGaO₄. They investigate with it the possible presence of a Berezinskii-Kosterlitz-Thouless phase (BKT) in this system.

Finding evidence of phases such a BKT which are difficult to probe is not a trivial task and the authors argue that TmMgGaO₄ evidences such a phase in a finite range of temperatures.

TmMgGaO₄ can be viewed as an effective spin-1/2 model with strong easy-axis anisotropy that can be mapped to a triangular lattice Ising model. Some of the authors had recently proposed in Nat.Comm. 10, 4530 (2019) that TmMgGaO₄ could bear a BKT phase making use of QMC and the exponential tensor renormalization group methods.

The present manuscript claims the observation of a BKT phase. The results are interesting but there are many issues that remain unclear:

1) Fig. 1 contains a summary of the measurements performed.

Based on the NMR observations (Fig. 2 c), the BKT phase is proposed to be confined at temperatures between 0.9K and 1.9K at zero out-of-plane magnetic fields. Below 0.9K the system shows long range magnetic order.

The question is, which phase/s are there at finite out-of plane magnetic fields?. Up to which finite magnetic fields does the assumed BKT survive? Hints of that seem to be given in Fig.3b in a very indirect way.

In other words, the manuscript presents nice data at finite out-of-plane fields but very little discussion on them.

2) The central result of the manuscript related to the statement of 'evidence of a BKT phase' is displayed in Fig.2c, which shows a plateau-like feature in the NMR spin-lattice relaxation rate as a function of temperature between 0.9K and 1.9K. Only results for in-plane fields of 3T are displayed, unclear is what the data at 1T show for this range of temperatures. Since this is the main result of the manuscript, could the authors show data for other in-plane fields? could the authors comment on the deviations between theory and experiment at temperatures above 1.9K (Fi.2c and Fig 2d)?

3) If I were given only Fig. 3 a and b, I would conclude that as a function of out-of-plane magnetic field and temperature the authors observe the transition to a long-range magnetic order at the smaller fields μH_1 which has an interesting field dependence before the phase undergoes a second phase transition at μH_2 , but there is no clear manifestation of a BKT transition in this figure.

Why don't the authors analyze and discuss these data, with the finite-field intermediate phase?

It is not clear how evidence of a BKT phase is seen here.

In Suppl. Fig1 the authors perform NMR measurements for out-of-field orientation that are useful for detecting the magnetic phase with long-range order, what about the BKT phase?

4) Fig. 1 shows the maxima of the specific heat data, could the authors show the data and whether they find signatures of the BKT phase they suggest?

The manuscript has potential but is not suitable for publication in Nat.Comm. in the present form.

Reviewer #2:

Remarks to the Author:

The manuscript presents the first NMR data on the highly topical TmMgGaO₄ spin compound, which is apparently well described by the triangular-lattice Ising Hamiltonian in an intrinsic weak transverse magnetic field, and is thus expected to present the Berezinskii-Kosterlitz-Thouless (BKT) phase above the low-temperature ordered phase. In particular, there are two recent publications on the subject/compound in Nat. Commun. [18,19] and one in Phys. Rev. X [16]. The manuscript brings very valuable microscopical characterization of the putative BKT phase, in particular as regards the low-energy spin fluctuations observed through NMR 1/T₁ relaxation rate, providing prominent characterization of the phase. The experimental data are novel, sound and important, and will eventually guarantee a publication. However, the data presentation and analysis are not optimal and should be improved by considering the following remarks:

1) Figs. 1 and 3: While the sharp features at *low*-field magnetization data denoted by arrows in Fig 3b have been reported as points in the phase diagram given in Fig. 1, the corresponding/symmetrical sharp features at *high* field are ignored. Please add these points to complete the given phase diagram.

2) The manuscript never mentions that TmMgGaO₄ compound presents important disorder induced by the Mg/Ga site mixing, although the importance of this disorder is apparently not settled in the literature (Ref. [16] vs Ref. [18]) and may be important for understanding the physics. The point is that NMR spectra are particularly sensitive to disorder, and the broadening of spectra on decreasing temperature presented in Fig. 2a is a typical NMR signature of some disorder.

3) The temperature dependence of the spectra shown in Fig. 2a is currently quantified only by calculating the first moment of the spectra to define their hyperfine shift (i.e., the average position), shown in Fig. 2b. Equally important information is to quantify the broadening of the spectra by calculating their 2nd moment, as well as to define the growth of the asymmetry of the spectrum by calculating the 3rd moment. The 2nd moment provides information on the effects of disorder, while both the 2nd and the 3rd moment should be discussed as a measure of the (average) order parameter of the low-temperature phase, a crucial quantity for characterizing/understanding the phase diagram.

4) The most valuable contribution of this work is the temperature dependence of the 1/T₁ NMR data (Fig. 2c) compared to the corresponding theoretical prediction by quantum Monte Carlo (QMC) given in Fig. 2d and Suppl. Fig. 2. The authors rightfully put forward important similarities between the experiment and theory: the high "plateau" of intense spin fluctuations in the BKT phase between the lower (T_L) and the upper (T_U) transition temperature, followed by a strong suppression of fluctuations in the ordered phase below T_L. By the way, showing a log-log plot of these latter data would be instructive.

However, the authors did not discuss equally important *dissimilarities* between the experiment and the theory: between 10 K and T_U the 1/T₁ data strongly *decrease* with decreasing temperature, and present a conspicuous step at T_U, which is in clear contradiction to the QMC predictions for the K point (in q-space) that supposedly provides the dominant contribution to 1/T₁: the prediction strongly *increases* with lowering temperature and does not present a noticeable step at T_U. Even if we suppose that above T_U the M-point data are dominant (Suppl. Fig. 2, see also below the remark 5), they do not present a step either, and are quite flat between 4 and 6 K, where the experimental data strongly vary.

In short, at and above T_U the NMR data are very much different from the QMC predictions,

meaning that either the employed model or the analysis (or both of them) are missing something important for understanding the involved physics. This should stimulate further work and should thus be clearly stated.

5) In any NMR investigation it is important to discuss the symmetry properties of the employed nuclear site (here Ga), to define the corresponding q-dependence of the hyperfine coupling tensor and the consequences for the data interpretation. This is not mentioned in the manuscript nor in the Supplementary Information (SI), although the information is crucial in discussing the contributions to $1/T_1$ from the K- and M-point in q-space ($1/T_1$ being the sum over all the q values). In Suppl. Fig. 2. that present the QMC predictions, we see that the (left axis) scale of the K-point contribution is more than an order of magnitude larger than the (right axis) scale of the M-point contribution. A straightforward conclusion would thus be that the latter contribution is always negligible, making the presented discussion of the relative competition of the two components as a function of temperature pointless.

Only the q-dependence of the Ga hyperfine coupling tensor could provide the "filtering" correction that could potentially make the two contributions comparable. Indeed, the Ga site seems to be symmetrically positioned with respect to the three nearest neighbouring Tm spins (forming a triangle), which certainly provides a specific q-dependence of the hyperfine coupling, which should be discussed (at least in the SI) in order to validate the proposed interpretation.

6) Minor details

a) Term "Knight-shift" is used for metallic systems only, while for spin systems we rather use "hyperfine shift".

b) The Curie-Weiss function is typically applied to the static (real part of) magnetic susceptibility, while $1/(T_1 \cdot T)$ provides information on dynamic (imaginary part of) susceptibility, which is quite different. Some further discussion/justification/explanations of the fits employed in SI would thus be helpful.

c) Please, align the zero levels of the left and the right scale of Suppl. Fig. 2.

d) Check for typos:

"... to solidify the finding ..." -> "... to strengthen the finding ..."

"experiment data" -> "experimental data"

Reviewer #3:

Remarks to the Author:

Dear Nature Communication Editor

I carefully read the manuscript by Hu et al. (255508_1_merged_1591600731). Although NMR in dilution refrigerator temperatures are always impressive, and, as far as I am concerned, impressive data is enough for publication regardless of the claims that follow, this time I found some serious difficulties with the manuscript.

The authors claim is that the system under investigation is Ising with no coupling to the in plane field required for the NMR experiment. But Ising systems do not have BTK transition. To fix this the authors say that above a certain temperature the system is not perfect Ising and have xy components. But if it has xy components they must be coupled to the applied in plane field required for the NMR experiment. Something is inconsistent here.

I have no idea what higher and lower BTK transition temperatures are. I thought that there is only one critical temperature in BTK model where vortices proliferate.

The authors claim that the broadening of the NMR lines is due to magnetic ordering. There are clearly two peaks in the data. Therefore, the broadening could also be due to lattice distortions. Can the two peak structure be understood by ordering only? Can the authors rule out lattice

distortions?

The main observation of the experimental work is a plateau in $1/T_1$ measurement. T_1 measurements in magnetic material at low temperature are tricky. The line is so broad that one does not excite the full line. The two peak structure could be due to different behavior of different spins belonging to different resonance frequency. The authors did not bother to check that their T_1 is H and H_1 independent along the line. The plateau could be simply due to a sum of two different contributions.

The agreement of the $1/T_1$ data with the numerical simulation is poor. There is no "quasi-plateau" in these simulations. In fact, there is no plateau at all. One can use the simulations to argue against a BTK transition in the system.

As for the magnetization measurements. I believe that the authors call M/H susceptibility and dM/dH differential susceptibility. We are not provided the raw of M vs. H data, but I believe there is no "peculiar magnetic correlations". If at low temperatures, M vs. H follows roughly the Brillouin function, then M/H will decrease with increasing H , and that is what we see in the data. At high enough temperatures M/H is H independent in all magnets.

In Fig. 3b there are two peaks. The high field one is called "quantum phase transition" but the low field one is not. It is not clear why, and between which two phases the "quantum phase transition" occurs?

The scaling analysis is not satisfactory. Usually, for scaling to be convincing one needs two order of magnitude variation in the scaled variable. The authors choose a field range of 0.6 to 0.9 T and M/H changes from 10 to 8. They could have chosen different field range and argue, equally well, that the scaling does not work.

Finally, I simply cannot see the connection between the Hamiltonian simulated in the manuscript and BTK transition. As mentioned above, there is no BTK transition in an Ising system, even with an internal field in the x direction. Take a look at Eq. 2 in the first paper I found by typing BTK transition in Google <https://www.phas.ubc.ca/~berciu/TEACHING/PHYS502/PROJECTS/18BKT.pdf>. Where are the phases required for BTK transition in the Hamiltonian of Eq. 1 in the manuscript?

To summarize, the experiments are quite standard, not done carefully enough, the comparison with theory is not satisfactory, the theory is not relevant, and the claims in the manuscript are not substantiated. Therefore, I do not support publication of this manuscript in Nature Communication

Manuscript NCOMMS-20-17512A-Z

Title: Evidence of the Berezinskii-Kosterlitz-Thouless Phase in a Frustrated Magnet

We thank the Reviewers' time and efforts very much for reviewing our manuscript. We find the comments valuable and insightful and have taken it as an excellent opportunity for us to further improve our work. In the following, we first summarize our response to all three Reviewers, and then give a point-by-point response to all the comments, where the text of Reviewers are cited in blue, followed by our subsequent response in the normal format. We have made substantial revisions in the figures, data analysis, and related discussions, in both the main body and the Supplementary Information. A Summary of Changes is appended after our detailed response, and the text changes are highlighted in the revised manuscript. We hope with these, the Reviewers will find the revised manuscript satisfactory and recommend it for publication.

Reviewer #1 stated that “finding evidence of phases such as BKT which are difficult to probe is not a trivial task”, and our work “claims the observation of a BKT phase”. He/she considered our results interesting, although at the same time raised a number of comments/questions. Reviewer #2 praised our work ‘brings very valuable microscopical characterization of the putative BKT phase’ and “provides prominent characterization of the phase”. He/she pointed out that our experimental data are “novel, sound and important, and will eventually guarantee a publication”. We thank the two Reviewers for the positive assessment and also their insightful comments on the experimental data and analyses, which we have carefully considered, responded to, and revised according to in our manuscript.

Reviewer #3 raised his/her main concern on the existence of a BKT phase in the triangular-lattice quantum Ising (TLI) model, together with some other questions. We think the concern of Reviewer #3 actually stand as a very representative misperception. It signifies that many people were not aware of the existence of BKT phase in a TLI model due to frustration and quantum fluctuations – this is partly due to the fact that such knowledge is mostly spread among the theoretical community of quantum magnetism. In fact, the emergence of a floating BKT phase in a TLI model has been fully established theoretically and very well documented in the literature for more than two decades, including the seminal work by Isakov and Moessner [Phys. Rev. B 68, 104409 (2003), Ref. 21 of the manuscript], by Wang et al where the emergent U(1) symmetry at the quantum critical point is explicitly shown [Phys. Rev. B 96.115160 (2017), Ref. 30], by some of us [Nature Commun. 11, 1111 (2020), Ref. 19], and a number of others. Now that our experimental results firmly realize such an intriguing BKT phase in a real material TmMgGaO_4 and with this, we hope an even larger community of physicists including experimentalists will be more familiar with such a fact. It is in this sense that we think his/her main concern actually speaks out exactly the novelty of this work. Nevertheless, we have now provided a more detailed introduction to the statistical theory on BKT phase transition in the TLI model, pointing out the emergent XY degree of freedom in the effective Hamiltonian, in the revised manuscript. We have also addressed his/her technical questions in the point-to-point response and made changes accordingly in the revised manuscript.

As appreciated by all the three Reviewers, the quasi-plateau structure in the $1/^{69}\text{T}_1$ measurement was the

FIG. R1: The QMC results of the spin-lattice relaxation rate $1/T_1$ vs. temperature T , where the contributions from all momentum \mathbf{q} points are included, weighted with the hyperfine form factor $|A_{\text{hf}}(\mathbf{q})|^2$ (c.f. Eq. R1 and Fig. R5).

main result of our work. In addition to the similarity between experimental data and numerical calculations, they also commented on some “dissimilarities”, and recommended further calculations to understand the deviation, which we highly appreciate and would like to make reply to immediately below.

The previous numerical data in Fig. 3d only included the spectral weight in the vicinity of K point in the momentum space, without contributions from other momentum points. As suggested by Reviewer #2, we now sum over $1/T_1$ for all \mathbf{q} points, weighted with the proper form factor $|A_{\text{hf}}(\mathbf{q})|^2$, and find a much better agreement between the theoretical and experimental data (see Fig. R1), both with a quasi-plateau feature. In particular, the measured $1/^{69}\text{Tl}$ at high- T , which increases with temperature, can also be captured by the dynamical quantum Monte Carlo TLI model calculations now.

Based on these further analysis, we would like to stress that our model effectively describes the real material in the sense that the essential physics in the material, such as the BKT fluctuations and their representation in the dynamical magnetic responses, are well captured. Needless to say, the real material is always more complicated. For example, influences from higher CEF levels above the non-Kramers doublet, the interlayer couplings not included in our model calculations, and the lack of precise knowledge on the form factor, etc, may explain the subtle difference still remained between the panels (c) and (d) of the revised Fig. 2 in the manuscript. But overall we hope that the Reviewers can now agree that the agreement between the experiment and theory is firm and convincing.

In the following, we give a point-by-point response to the comments of all three Reviewers.

Response to the first Reviewer's report

Reviewer #1 : *The authors present a combination of NMR, susceptibility and specific heat data with QM-C simulations for the two-dimensional antiferromagnet TmMgGaO₄. They investigate with it the possible presence of a Berezinskii-Kosterlitz-Thouless phase (BKT) in this system.*

Finding evidence of phases such a BKT which are difficult to probe is not a trivial task and the authors argue that TmMgGaO₄ evidences such a phase in a finite range of temperatures.

TmMgGaO₄ can be viewed as an effective spin-1/2 model with strong easy-axis anisotropy that can be mapped to a triangular lattice Ising model. Some of the authors had recently proposed in [Nat. Comm. 10, 4530 (2019)] that TmMgGaO₄ could bear a BKT phase making use of QMC and the exponential tensor renormalization group methods.

Reply: We thank the reviewer for his/her positive assessment of our work. As proposed theoretically in Ref. 19 by some of us, BKT transitions occur in the frustrated quantum magnet TmMgGaO₄, which, however, should exhibit no conventional thermodynamic singularity and thus challenging to be confirmed experimentally. Therefore, the NMR measurements are employed to provide a dynamical detection of the BKT transition in this magnet. We are excited to find the low-energy fluctuations and dissipations in TmMgGaO₄ can be sensitively probed by $1/^{69}T_1$, from which clear NMR signature of the BKT phase is observed. The experimental data are found to be in great agreement with the QMC calculations, constituting a hallmark of the BKT transition in the compound.

Reviewer #1 : *The present manuscript claims the observation of a BKT phase. The results are interesting but there are many issues that remain unclear:*

1) Fig. 1 contains a summary of the measurements performed. Based on the NMR observations (Fig. 2 c), the BKT phase is proposed to be confined at temperatures between 0.9K and 1.9K at zero out-of-plane magnetic fields. Below 0.9K the system shows long range magnetic order.

The question is, which phase/s are there at finite out-of-plane magnetic fields? Up to which finite magnetic fields does the assumed BKT survive? Hints of that seem to be given in Fig.3b in a very indirect way.

In other words, the manuscript presents nice data at finite out-of-plane fields but very little discussion on them.

Reply: In this work we mainly focus on the zero-field case and study the BKT phase transitions therein. From theoretical calculations based on the TLI model, believed to describe the material TmMgGaO₄, the floating BKT phase is “fragile” and cannot survive finite out-of-plane fields, since the latter constitutes a relevant perturbation to the BKT phase. Nevertheless, through analyzing the scaling behaviors of the magnetization data, we still find a regime near the BKT phase with remnant of BKT fluctuations at small out-of-plane fields.

When the out-of-plane fields are large, the nature of the ordered phase under the “dome” in Fig. 1 of the manuscript is a very interesting, but is still an open question we are currently working on. In the present

FIG. R2: The spin-spin relaxation rate $1/^{69}T_2$ as functions of temperatures with different in-plane and out-of-plane fields.

work, we analyze the susceptibility data at finite out-of-plane fields in Fig. 3 to reveal the asymptotical scaling behaviors related to the zero-field BKT phase. We refer the Reviewer to our detailed answers to his/her comment #3) on the same aspect.

Reviewer #1 : 2) *The central result of the manuscript related to the statement of ‘evidence of a BKT phase’ is displayed in Fig.2c, which shows a plateau-like feature in the NMR spin-lattice relaxation rate as a function of temperature between 0.9K and 1.9K. Only results for in-plane fields of 3T are displayed, unclear is what the data at 1T show for this range of temperatures. Since this is the main result of the manuscript, could the authors show data for other in-plane fields?*

Reply: We thank the reviewer very much for this insightful question. The NMR measurements with in-plane fields smaller than 3 T and with temperatures below 1.8 K are unfortunately not available due to technical limitations. In fact, we tried very hard for these fields, but cannot manage to obtain meaningful data of $1/^{69}T_1$. This is mainly because the echo intensity is found to be greatly reduced in the low-field and low-temperature regime, as we elaborate in details below.

First, the spectral weight is reduced to 1/9 when the applied field is reduced from 3 T to 1 T (empirically, the total spectra weight $\propto H^2$). Second, the measured spin-spin relaxation $1/^{69}T_2$ rises up quickly when cooled down below 4 K and increases significantly with decreasing field (see Fig. R2 and also Suppl. Fig. 5 in the Suppl. Information). At 1.9 K, the T_2 is reduced to about 50 μs at 1 T. The delay time between two echo pulses is set as 20 μs at 1 T to reduce the large RF ring at such low frequencies, which results in an echo signal inductions by 55% ($e^{-40/50} \approx 0.45$). Below 1.9 K, the reduction would grow even more rapidly, indicated by the strong upturn in $1/^{69}T_2$ in Fig. R2. Due to the above two effects, we are unable to perform $1/^{69}T_1$ measurements below 1.9 K with fields of 1 and 2 T, as the signal to noise ratio is too small.

Whereas for in-plane fields of 4 T and higher, the sample cannot be held in position because of the large

anisotropy in the g factor and unavoidable sample misalignment (less than 2°).

In the Methods part of the revised manuscript, we have explained why only the 3-T data are shown.

Reviewer #1 : *Could the authors comment on the deviations between theory and experiment at temperatures above 1.9 K (Fig. 2c and Fig. 2d)?*

Reply: We thank the reviewer for this important question. Previously, we only included the K' point (a momentum point in the close vicinity of K point) contribution in Fig. 2d. We now recalculate the $1/T_1$ by including all q contributions in the Brillouin zone, with the hyperfine coupling form factor considered, and find much better agreement with the experiment. In particular, the results show that the contributions from points other than K' are significant at temperatures above ~ 2 K. We kindly refer the Reviewer to Fig. R1 and our detailed explanation in the introductory part of our reply.

With this, we would conclude and further stress that the enhanced fluctuations in the BKT phase are present in both experiment and model calculations.

Reviewer #1 : *3) If I were given only Fig. 3a and b, I would conclude that as a function of out-of-plane magnetic field and temperature the authors observe the transition to a long-range magnetic order at the smaller fields $\mu H1$ which has an interesting field dependence before the phase undergoes a second phase transition at $\mu H2$, but there is no clear manifestation of a BKT transition in this figure.*

Why don't the authors analyze and discuss these data, with the finite-field intermediate phase?

It is not clear how evidence of a BKT phase is seen here.

Reply: We thank the referee for this valuable question. In this work, we focus mostly on the BKT phase at zero field, and it is known from the field theory that the BKT phase is fragile under finite out-of-plane fields. Therefore, we do not expect a clear BKT transition at finite fields. Please see also our response to comment #1) above.

Nevertheless, the nature of intermediate-field phase also constitutes a very interesting question. From the current measurements and calculations, we find it is a magnetically ordered phase and mentioned it in the caption of Fig. 1 in the revised manuscript to address the Reviewer's question. However, so far we cannot tell whether it is an AF clock or a ferrimagnetic phase based on the data at hand. To fully clarify the nature of this ordered phase, more experimental measurements and theoretical calculations are needed. We are currently working along this line, and will report more results in a separate paper when a definite conclusion can be drawn. We have added discussions on the finite-field aspects in the revised manuscript.

Reviewer #1 : *In Suppl. Fig. 1 the authors perform NMR measurements for out-of-field orientation that are useful for detecting the magnetic phase with long-range order, what about the BKT phase?*

Reply: In fact, we did not measure the long-range ordered phase directly. The $1/^{69}T_1$ data shown in Suppl. Fig. 1 is measured down to 1.5 K, and the transition temperature is obtained via data extrapolation, which is consistent with other measurements, including the specific heat and magnetization curves, etc. Because of the short $^{69}T_2$ with out-of-plane field as demonstrated in Fig. R2, the $1/^{69}T_1$ measurements are

FIG. R3: The specific heat curves C_m/T of TmMgGaO_4 measured under various magnetic fields, taken from Y. Li, *et al.*, PRX 2020 (Ref. 16 of the manuscript). Below ~ 300 mK, C_m/T diverges due to the nuclear spins.

not accessible below 1.5 K as well. Furthermore, as we explained above, the BKT phase shall not survive at finite out-of-plane fields, although fluctuations may still remain at small fields. However, the NMR signal is too weak for precise $1/^{69}\text{T}_1$ measurements to detect those remnant BKT fluctuations below 1.5 K and 1 T, as explained above in our reply to point #2). In this work, we mainly focus on the BKT phase under zero field, and leave the detailed investigation of the case with finite out-of-plane fields in a future study [see also our reply above to point #3)].

Reviewer #1 : 4) Fig. 1 shows the maxima of the specific heat data, could the authors show the data and whether they find signatures of the BKT phase they suggest?

Reply: As mentioned in the caption of Fig. 1, the contour background depicts the magnetic specific heat data under various fields were adapted from Fig. 8(b) of Ref. 16 (Fig. R3). We find signatures of order-disorder transition under finite out-of-plane fields, evidenced by a rather sharp peak in the of C_m/T curves (the maximal position collected and plotted in Fig. 1 of the manuscript). However, since the BKT phase transition is an infinite-order transition, one *cannot* observe direct signature of the BKT transition in the zero-field specific heat curve (see the 0-T curve in Fig. R3).

Reviewer #1 : *The manuscript has potential but is not suitable for publication in Nat.Comm. in the present form.*

Reply: We thank the Reviewer for the valuable comments, the answer to which are important to improve the manuscript and to lay out open questions on finite field properties. In the resubmitted manuscript, we provide evidence why the window for the NMR measurements of the BKT phase is narrow with the supplementary $1/^{69}\text{T}_2$ data, and a much better comparison to experimental results by summing over q points in the calculations. With these improvements, we hope that the Reviewer can agree that our work is now suitable for publication in Nature Communications.

Response to the second Reviewer's report

Reviewer #2 : *The manuscript presents the first NMR data on the highly topical $TmMgGaO_4$ spin compound, which is apparently well described by the triangular-lattice Ising Hamiltonian in an intrinsic weak transverse magnetic field, and is thus expected to present the Berezinskii-Kosterlitz-Thouless (BKT) phase above the low-temperature ordered phase. In particular, there are two recent publications on the subject/compound in *Nat. Commun.* [18, 19] and one in *Phys. Rev. X* [16]. The manuscript brings very valuable microscopical characterization of the putative BKT phase, in particular as regards the low-energy spin fluctuations observed through NMR $1/T_1$ relaxation rate, providing prominent characterization of the phase. The experimental data are novel, sound and important, and will eventually guarantee a publication.*

Reply: We thank the Reviewer for the very positive evaluation of our work, as well as his/her very insightful and constructive comments/suggestions below.

Reviewer #2 : *However, the data presentation and analysis are not optimal and should be improved by considering the following remarks:*

1) Figs. 1 and 3: While the sharp features at 'low'-field magnetization data denoted by arrows in Fig 3b have been reported as points in the phase diagram given in Fig. 1, the corresponding/symmetrical sharp features at 'high' field are ignored. Please add these points to complete the given phase diagram.

Reply: We thank the second Referee for pointing out this, which is indeed a very nice suggestion. We have added these points in the updated Fig. 1, and found them fall into the same dome-shape line, together with points from other experimental measurements.

Reviewer #2 : *2) The manuscript never mentions that $TmMgGaO_4$ compound presents important disorder induced by the Mg/Ga site mixing, although the importance of this disorder is apparently not settled in the literature (Ref. [16] vs Ref. [18]) and may be important for understanding the physics. The point is that NMR spectra are particularly sensitive to disorder; and the broadening of spectra on decreasing temperature presented in Fig. 2a is a typical NMR signature of some disorder.*

Reply: We thank the Reviewer for raising the questions on the disorder effects in the NMR spectra due to the Mg/Ga site mixing. Following the nice suggestions here and also in point #3) below, we have computed the second moment of the spectra, i.e., the standard deviation of intensity distribution that characterizes the peak broadening and roughly equals the full-width at half-maximum (FWHM), and now include the results in Fig. R4 (and also in Suppl. Fig. 4).

The second moment is about 0.85 MHz at $T=10$ K, much larger than the $1/^{69}T_2$ (4 kHz) under the same field (see Fig. R2), indicating an inhomogeneous broadening of the spectra. At 5 K and above, the linewidth does not change much with temperature (cf. the data at 5 K and 10 K), nor with field (cf. the results under 1 T and 3 T in Fig. R4), indicating that the high-temperature broadening does not correspond to a magnetic origin. Rather, since ^{69}Ga has a 3/2 nuclear spin, its quadrupolar moment is coupled to the local electric-field-gradient (EFG), whose first-order correction to the resonance frequency barely changes with fields and

FIG. R4: The second and the third moments of the NMR spectra as functions of temperatures, under 1 T and 3 T in the ab -plane.

temperatures in principle. Therefore, the high-temperature broadening of the spectra indicates strong EFG inhomogeneity, likely due to the Mg/Ga site mixing in the compound.

Below 2 K, on the other hand, the dramatic broadening of the NMR linewidth upon cooling should be caused by the inhomogeneity of the local hyperfine field, as it occurs when the static/quasi-static magnetic ordering develops. The frozen Mg/Ga site mixing ought to also enhance this inhomogeneity, since the hyperfine field and/or the dipolar field produced by the Tm^{3+} magnetic ions on the ^{69}Ga nucleus is affected by its exact position and the neighboring ions. However, it remains an open question to which extent the local moment of the ^{69}Ga , and consequently the magnetic properties, are affected by the site mixing. This is in particular thought-provoking, as the thermodynamic properties of TmMgGaO_4 can be well fitted by the TLI model without randomness, the magnetic order phase is maintained at low temperatures, and the spin-wave dispersion remains sharp as observed in INS measurements (see Ref. 19).

We have added some discussions on the possible influence of randomness in the main text of the revised manuscript as well as in Suppl. Note 4.

Reviewer #2 : 3) *The temperature dependence of the spectra shown in Fig. 2a is currently quantified only by calculating the first moment of the spectra to define their hyperfine shift (i.e., the average position), shown in Fig. 2b. Equally important information is to quantify the broadening of the spectra by calculating their 2nd moment, as well as to define the growth of the asymmetry of the spectrum by calculating the 3rd moment. The 2nd moment provides information on the effects of disorder, while both the 2nd and the 3rd moment should be discussed as a measure of the (average) order parameter of the low-temperature phase, a crucial quantity for characterizing/understanding the phase diagram.*

Reply: This is a very constructive comment. The second moment is discussed in our reply to comment #2), where the high-temperature line broadening can be associated with structural disorder from site mixing,

and the additional low temperature line broadening, on the contrary, is believed to reflect the inhomogeneous hyperfine field.

The third moment, representing asymmetry of the spectra, is also plotted vs. T in Fig. R4 under the magnetic field of 3 T. The third moment changes its sign from negative to positive when cooled below 2 K, suggesting the formation of static or quasi-static magnetic ordering. Since NMR is a precise low-energy probe, $1/^{69}\text{Tl}$ measures the low-energy dynamics directly, and Fig. R4 shows that the second and third moments detect the static order (below T_N) and the quasi-static order (in the BKT phase) very sensitively. These comments are added in the revised main text and in the Suppl. Note 4.

Reviewer #2 : *4) The most valuable contribution of this work is the temperature dependence of the $1/T1$ NMR data (Fig. 2c) compared to the corresponding theoretical prediction by quantum Monte Carlo (QMC) given in Fig. 2d and Suppl. Fig. 2. The authors rightfully put forward important similarities between the experiment and theory: the high “plateau” of intense spin fluctuations in the BKT phase between the lower (T_L) and the upper (T_U) transition temperature, followed by a strong suppression of fluctuations in the ordered phase below T_L . By the way, showing a log-log plot of these latter data would be instructive.*

However, the authors did not discuss equally important ‘dissimilarities’ between the experiment and the theory: between 10 K and T_U the $1/T1$ data strongly ‘decrease’ with decreasing temperature, and present a conspicuous step at T_U , which is in clear contradiction to the QMC predictions for the K point (in q -space) that supposedly provides the dominant contribution to $1/T1$: the prediction strongly ‘increases’ with lowering temperature and does not present a noticeable step at T_U . Even if we suppose that above T_U the M-point data are dominant (Suppl. Fig. 2, see also below the remark 5), they do not present a step either, and are quite flat between 4 and 6 K, where the experimental data strongly vary.

In short, at and above T_U the NMR data are very much different from the QMC predictions, meaning that either the employed model or the analysis (or both of them) are missing something important for understanding the involved physics. This should stimulate further work and should thus be clearly stated.

Reply: Thanks for this comment, which urges us to reconsider our theoretical analysis. Interestingly, by including contributions from all q points, we indeed can observe the plateau-like structure at intermediate temperature and accordant tendency above the higher temperature scale T_U^* , as shown in Fig. R1. Although the material is always more complicated than the simplified model, we believe our model effectively describes the real material, and the experimental measurements and model calculations consistently confirm the presence of BKT physics in TmMgGaO_4 . We kindly refer the Reviewer to the introductory part of the reply for more detailed discussions.

Reviewer #2 : *5) In any NMR investigation it is important to discuss the symmetry properties of the employed nuclear site (here Ga), to define the corresponding q -dependence of the hyperfine coupling tensor and the consequences for the data interpretation. This is not mentioned in the manuscript nor in the Suppl. Information (SI), although the information is crucial in discussing the contributions to $1/T1$ from the K- and M-point in q -space ($1/T1$ being the sum over all the q values). In Suppl. Fig. 2. that present the QMC*

FIG. R5: (a) Lattice structure within the triangular plane where each Ga ion has three nearest Tm sites. (b) shows the contour plot of corresponding form factor $|A_{\text{hf}}(\mathbf{q})|^2$.

predictions, we see that the (left axis) scale of the K-point contribution is more than an order of magnitude larger than the (right axis) scale of the M-point contribution. A straightforward conclusion would thus be that the latter contribution is always negligible, making the presented discussion of the relative competition of the two components as a function of temperature pointless.

Only the q -dependence of the Ga hyperfine coupling tensor could provide the filtering correction that could potentially make the two contributions comparable. Indeed, the Ga site seems to be symmetrically positioned with respect to the three nearest neighbouring Tm spins (forming a triangle), which certainly provides a specific q -dependence of the hyperfine coupling, which should be discussed (at least in the SI) in order to validate the proposed interpretation.

Reply: Thanks a lot for this very constructive comment, which is also related to comment #4). In order to include all \mathbf{q} contributions in $1/^{69}\text{T}_1$, we consider the form factor below,

$$A_{\text{hf}}(\mathbf{q}) = \sum_j \tilde{A}_{\text{hf}} e^{i\mathbf{q}(r_j - r_i)}, \quad (\text{R1})$$

where $r_{i(j)}$ labels the position of Ga(Tm) ions, and j is NN sites of site i (see Fig. R5). By assuming a constant \tilde{A}_{hf} , we compute the form factor $A_{\text{hf}}(\mathbf{q})$ and show the results in Fig. R5(b), from which we observe that indeed the K point has a very small form factor (actually zero for isotropic \tilde{A}_{hf}) while the M point corresponds to a larger value.

Besides the form factor $A_{\text{hf}}(\mathbf{q})$, the difference in magnetic density of states are also very different between K and M points. The lattice model has a quadratic dispersion and thus has a much larger number of \mathbf{q} points around the M point than around the K point. The difference, as checked, is by one order of magnitude. In all, we include the hyperfine coupling form factor $A_{\text{hf}}(\mathbf{q})$ and sum over all \mathbf{q} points, and obtain $1/^{69}\text{T}_1$ results which are in a much better agreement with experimental results (cf., Fig. R1).

Reviewer #2 : 6) Minor details:

a) Term “Knight-shift” is used for metallic systems only, while for spin systems we rather use “hyperfine shift”.

b) The Currie-Weiss function is typically applied to the static (real part of) magnetic susceptibility, while $1/(T_1T)$ provides information on dynamic (imaginary part of) susceptibility, which is quite different. Some further discussion/justification/explanations of the fits employed in SI would thus be helpful.

c) Please, align the zero levels of the left and the right scale of Suppl. Fig. 2.

d) Check for typos: “to solidify the finding” → “to strengthen the finding”; “experiment data” → “experimental data”.

Reply: Thanks for pointing out them to us, and we have now made corresponding revision in the resubmitted manuscript.

Response to the third Reviewer's report

Reviewer #3 : *I carefully read the manuscript by Hu et al. (255508_1_merged_1591600731). Although NMR in dilution refrigerator temperatures are always impressive, and, as far as I am concerned, impressive data is enough for publication regardless of the claims that follow, this time I found some serious difficulties with the manuscript.*

The authors claim is that the system under investigation is Ising with no coupling to the in plane field required for the NMR experiment. But Ising systems do not have BTK transition. To fix this the authors say that above a certain temperature the system is not perfect Ising and have xy components. But if it has xy components they must be coupled to the applied in plane field required for the NMR experiment. Something is inconsistent here.

Reply: We thank the Reviewer to raise this question and believe that this is a simple misunderstanding and can be resolved with detailed explanation below.

The reviewer is correct that indeed, conventional 2D Ising models do not host any BKT transition. However, it does for a frustrated transverse-field Ising model on a triangular lattice (TLI). It has been well established theoretically that the model hosts a floating BKT phase enclosed by two successive BKT transitions. For the convenience of the Reviewer, we adapted the original phase diagram in Ref. 21 and show it as Fig. R6 in this reply. So to speak, the theoretical consensus of the BKT phase in TLI model is well-established. What is missing is its material realization and experimental detection in quantum Ising compounds. And that we think, is the major contribution of this work.

We thus feel excited for our finding in TmMgGaO_4 , and would like to share with the Reviewer that the “secrete” to realize a BKT phase in this Ising magnet right lies in the key ingredient of *magnetic frustration*. In the TLI model, there emerges effective XY degrees of freedom ψ at intermediate temperature, which is a combination of the Ising (Z) components $m_{A,B,C}^z$ on three sublattices, i.e.,

$$\psi = m_A^z + e^{i2\pi/3} m_B^z + e^{i4\pi/3} m_C^z. \quad (\text{R2})$$

Please note that $\psi = |\psi|e^{i\theta}$ is a complex order parameter, representing the emergent XY degree of freedom, which plays an essential role in the BKT phase transition. We have added a brief introduction to the concrete form of the emergent ψ in in the Methods part of our revised manuscript. For more details, please refer to our previous work (Ref. 19).

Reviewer #3 : *I have no idea what higher and lower BTK transition temperatures are. I thought that there is only one critical temperature in BTK model where vortices proliferate.*

Reply: In the TLI model, there exists a two-step melting of the magnetic clock order (cf. Fig. R6). Near the upper BKT transition, vortices formed by the XY degrees of freedom ψ proliferate. This well falls into the “critical temperature in BKT model where vortices proliferate” mentioned by the Reviewer.

The existence of a second (lower) BKT transition in this system is also very interesting. In field theory,

FIG. R6: Finite-temperature phase diagram of the TLI model, taken from Isakov and Moessner, Phys. Rev. B 68, 104409 (2003) (Ref. 21 of the manuscript). The horizontal axis is the transverse field Γ , and the vertical one temperature T , both in the unit of Ising coupling J . The data points collapse into two lines represent the two (upper and lower) BKT phase transitions, enclosing the floating BKT phase. Besides, there also include the low-temperature clock ordered and the high-temperature paramagnetic phases.

the TLI model behaves very much like a XY model at an intermediate temperature (i.e., in the floating BKT phase), except for the existence of a \mathbb{Z}_6 term in the effective Hamiltonian. This term can eventually lead to a \mathbb{Z}_6 symmetry breaking in the clock-ordered AF phase at low temperatures. Consequently, there occurs a second BKT transition T_L to separate the BKT phase and clock phase. The lower BKT transition at T_L is actually dual to the upper one at T_U , and this scenario of two BKT transitions has been very well established in previous theoretical studies of the TLI model. We refer the Reviewer to Refs. 19–21, 30 of the manuscript for more details.

Reviewer #3 : *The authors claim that the broadening of the NMR lines is due to magnetic ordering. There are clearly two peaks in the data. Therefore, the broadening could also be due to lattice distortions. Can the two peak structure be understood by ordering only? Can the authors rule out lattice distortions?*

Reply: For clarification, we have now analyzed the second moment (linewidth) and the third moment (spectrum asymmetry) of the measured NMR spectra, and plotted the results in Fig. R4. Below we recapitulate the discussions, and refer the Reviewer to our detailed reply to comment points #2) and 3) of Reviewer #2 on the NMR spectra analysis, and also Suppl. Note 4.

The broadening of NMR lines above 2 K barely varies with fields and temperatures, which indeed constitutes an experimental evidence for lattice disorder due to Mg/Ga site mixing. On the other hand, the dramatic change of the second and the third moments below 2 K is ascribed to the inhomogeneity of the local hyperfine field. This is fully consistent with the onset of quasi-static magnetic order below 2 K (and the onset of the static ordering below 1 K), as also observed in, e.g., the neutron scattering measurements

(cf. Refs. 16, 18).

Lastly, we do not think it is sensible to ascribe the line broadening below 2 K to further lattice distortion in the material. The lattice degrees of freedom should have been well frozen for solid-state materials at such low temperature. Indeed, there has been no structural transition reported below 2 K in the literature (cf. Refs. 16, 18).

Reviewer #3 : *The main observation of the experimental work is a plateau in $1/T_1$ measurement. T_1 measurements in magnetic material at low temperature are tricky. The line is so broad that one does not excite the full line. The two peak structure could be due to different behavior of different spins belonging to different resonance frequency. The authors did not bother to check that their T_1 is H and H_1 independent along the line. The plateau could be simply due to a sum of two different contributions.*

Reply: Thanks for the question. We did check the H and H_1 dependence of T_1 at one particular temperature, and have amended this information in the Methods part of the main text. To be specific, the frequency and the energy dependence of the $1/^{69}T_1$ data are examined at 1.2 K, which is also important to check the heating effect. We have tested H_1 with two different energies, 14 mT and 24 mT (corresponding to π -pulse of the 1.2 μ s and 2 μ s, respectively), yet see no difference in $1/^{69}T_1$ within the error bar. The $1/^{69}T_1$ results were also checked with different frequencies across the NMR line at 1.2 K, and no variation was found within the error bar. These observations clearly rule out any H and H_1 dependence of $1/^{69}T_1$ and the heating effect. With this, we fixed $1/^{69}T_1$ measurements right at the peak position of the spectra, with H_1 of 24 mT.

Reviewer #3 : *The agreement of the $1/T_1$ data with the numerical simulation is poor. There is no quasi-plateau in these simulations. In fact, there is no plateau at all. One can use the simulations to argue against a BTK transition in the system.*

Reply: We thank the reviewer for pointing this out, but we humbly disagree and would like to stress that the model simulation and the experimental observations are in fact quite similar. When comparing the model with experiments in particular in the strongly correlated systems, we need to bear in mind that the model is an approximation to the real material, and in this way, the model could only capture the essential physical features of the material.

Given that, also taking the other two Reviewers' comments into account, we have now summed over all q -point contributions and found significantly improved comparisons between numerical and experimental results. In particular, the plateau-like feature is much more obvious in the numerical data. This is already discussed in the introductory part of this reply, and we kindly refer the referee to our new data and related arguments there, as well as the updated Fig. 2d in the revised manuscript.

Reviewer #3 : *As for the magnetization measurements. I believe that the authors call M/H susceptibility and dM/dH differential susceptibility. We are not provided the raw of M vs. H data, but I believe there is no "peculiar magnetic correlations". If at low temperatures, M vs. H follows roughly the Brillouin function, then M/H will decrease with increasing H , and that is what we see in the data. At high enough temperatures*

M/H is H independent in all magnets.

Reply: We are sorry but firmly disagree with the Reviewer for interpreting our data as systems with no “peculiar magnetic correlations”. There exists, clearly, evidence for strong magnetic correlations in the system. To facilitate the discussion, we take the Reviewer’s suggestion and provide the M - H curve below in Fig. R7 (as well as in Suppl. Note 6).

As shown in Fig. R7, the M - H curves at low temperatures exhibit two inflection points at fields about 1.3 T and above 2.3 T, which is completely different from the paramagnetic Brillouin function (for non-interacting spin systems), as the dM/dH of the latter decreases monotonically as H increases. At very high temperatures, certainly M/H is H independent for a paramagnet. Indeed, from 10 K to 70 K, the M/H curves are observed to follow a universal Curie-Weiss law, with a fitted Curie temperature as large as -19 K (see, e.g., Ref. 18), which clearly indicates a strong antiferromagnetic interaction between spins. Please also be reminded that TmMgGaO_4 is a strongly correlated magnet, whose spin correlations are built up upon cooling. Through a BKT phase with algebraic spin order, the system eventually ends up with a long-range AF order at sufficiently low temperature (below about 1 K).

FIG. R7: The magnetization as functions of fields at selected temperatures. Data at different temperatures are shifted vertically for clarity.

Reviewer #3 : *In Fig. 3b there are two peaks. The high field one is called quantum phase transition but the low field one is not. It is not clear why, and between which two phases the quantum phase transition occurs?*

Reply: We thank the Reveiwer for this very good question. The ground state at zero field is anti-ferromagnetically ordered. As a Matter of fact, the quantum states at finite fields in TmMgGaO_4 (in the dome-like region in the phase diagram) constitutes a very interesting and intriguing question on its own. We are currently working to resolve its magnetic nature, and the possibilities include the clock antiferromagnetic, ferrimagnetic, and other types of magnetic order. Then the connection between the zero-field and

FIG. R8: Fitting of differential susceptibility at a much larger field range, where a $\alpha = 2/3$ scaling can be still seen clearly.

the finite-field phases becomes complicated. As Fig. 3b of the main text shows, the low-field kink in the dM/dH disappears at $T = 0.4$ K, with no indication of phase transitions, and therefore the existence of the quantum phase transition (at zero temperature) at the low field side becomes questionable. We hope to clarify the the quantum states and phase transitions under finite out-of-plane fields in a future study once the conclusion is drawn. In the present work, although we focus on the BKT phase at zero field, a sentence is added in the main text (right column, page 3) to briefly mention the question.

Reviewer #3 : *The scaling analysis is not satisfactory. Usually, for scaling to be convincing one needs two order of magnitude variation in the scaled variable. The authors choose a field range of 0.6 to 0.9 T and M/H changes from 10 to 8. They could have chosen different field range and argue, equally well, that the scaling does not work.*

Reply: In general, that is right. However, here we are dealing with BKT fluctuations which should be weak and may only show up in a narrow temperature and field window, especially when an ordered phase is nearby. In practice we indeed only find a limited temperature window to observe this scaling, in both experimental and simulated data. Nevertheless, we take the Reviewer's comment, extend the scaling to a bit larger field range ([0.6, 1.4] T), and find the scaling works equally well (see Fig. R8 attached). From theoretical side, the presence of the scaling in TLI system (cf. Ref. 23, 24), and the fitted exponent $\alpha = 2/3$ agrees well with the field theory, showing that this is *not* a coincidence but a strong evidence for the existence of BKT transition in the material.

Reviewer #3 : *Finally, I simply cannot see the connection between the Hamiltonian simulated in the manuscript and BTK transition. As mentioned above, there is no BTK transition in an Ising system, even with an internal field in the x direction. Take a look at Eq. 2 in the first paper I found by typing BTK transition in*

Google [<https://www.phas.ubc.ca/~berciu/TEACHING/PHYS502/PROJECTS/18BKT.pdf>.] *Where are the phases required for BTK transition in the Hamiltonian of Eq. 1 in the manuscript?*

Reply: The phase required for BKT transition lies in the emergent XY degree of freedom ψ , as indicated in Eq. (R2) above. Please see our explanation on the origin of the BKT transition there. For readers' convenience, we also added this definition of variable ψ in the Methods part of the revised manuscript.

Reviewer #3 : *To summarize, the experiments are quite standard, not done carefully enough, the comparison with theory is not satisfactory, the theory is not relevant, and the claims in the manuscript are not substantiated. Therefore, I do not support publication of this manuscript in Nature Communication.*

Reply: First of all, we think we have very well answered the questions and comments raised by the Reviewer, although some of his/her comments are unfortunately due to misunderstanding of our work. The theory is relevant. The existence of the floating BKT phase in the transverse-field Ising model on the triangular lattice has been known in theory since nearly two decades ago. The comparison between theory and experiment is now in great agreement in the revised version, by considering the influence of hyperfine form factor and including contributions from all \mathbf{q} points, which further substantiates our claim. Besides, the H and H_1 dependence of the $1/^{69}\text{T}_1$ is also carefully checked.

The experimental setup is by no means just a standard application of NMR techniques, but requires a careful design with insight into the material TmMgGaO_4 . The BKT phase is expected to be weak and may only be detectable in a narrow window range in principle, and our measurements employ a in-plane field which is strong enough to detect accurate NMR signals yet does not disturb the BKT phase itself. The Reviewer's confusion on the existence of the BKT phase in an "Ising system" *in fact rightly speaks the exact novelty of our experimental findings here* as our work demonstrates unambiguously that such an exotic phase not only exists in theory, but can also be found in a real material. Notably, Reviewers #1 and #2 both speak highly of our approach in both experiment and theory.

With our response above and our revisions to the manuscript, we sincerely hope the Reviewer #3 can be convinced that our work can be published in Nature Communications.

Summary of Changes

The summary of the major changes in the resubmitted text are shown below.

1. The phase diagram in Fig. 1 is updated with inclusion of higher characteristic field from the dM/dH curves. A sentence is added in the main text to note the magnetic state below the dome-like region is an open question.
2. The quantum Monte Carlo data in Fig. 2d is replaced, by summing over all q points in the BZ. Accordingly, we also revised the related discussions on right column, page 2 of the manuscript.
3. Fig. 3b is updated by adding arrows pointing out the higher characteristic field locations.
4. The possible influences of randomness, together with some other points, e.g., vortex fluctuations, are added in the discussion part.
5. The absence of the field and energy dependence of T_1 is noted in the Methods of the main text.
6. An introduction to the emergent U(1) symmetry is added in the Methods of the main text.
7. Supplementary Notes 2 and Fig. 2 are added to address the hyperfine coupling form factor.
8. Supplementary Notes 4 and Fig. 4 are added to discuss the second and third moments of NMR spectra. Related discussion is also included in the main text of the manuscript.
9. Supplementary Notes 5 and Fig. 5 are added to show the $1/^{69}T_2$ results.
10. Supplementary Notes 6 and Fig. 6 are added to provide $M-H$ data at various temperatures.
11. Supplementary Notes 7 and Fig. 7 are added to cover the scaling of differential susceptibilities in an extended range of fields.
12. We also corrected minor typos and misspellings in the revised manuscript.

Reviewers' Comments:

Reviewer #1:

Remarks to the Author:

In the revised version of the manuscript the authors took carefully account of the comments raised in my report and clarified a few open issues. I recommend the present version of the manuscript for publication in Nature Communications.

Reviewer #2:

Remarks to the Author:

In the revised manuscript, the authors have taken into account the referees' remarks, and have, in particular, considerably improved the analysis by:

A) adding the temperature dependence of the 2nd and the 3rd moment of the NMR spectra and by
B) taking into account the q-dependence of the hyperfine coupling tensor in the analysis of the $1/T_1$ data.

Notably, the later point B) improved the agreement of the theoretical prediction and measured NMR $1/T_1$ data (reflecting the low-energy spin fluctuations), so that the conclusions of the manuscript are now more convincing. However, regarding the modified parts of the manuscript I would have several comments and suggestion (given below) to improve the presentation. I would recommend the publication after the authors have taken them into account:

1) Regarding the information on the static/average spin distribution given by NMR spectra, from Fig. 2b and Supplementary Fig. 4, we learn that for the two putative BKT transitions the lower one at T_L is clearly visible **only** in the 1st moment, while the upper one at T_U is clearly visible **only** in the 3rd moment. This is a quite unexpected situation, which merits to be explicitly underlined/commented in the manuscript

2) The Fig. 2 caption should **explicitly** mention that the sum over q includes the hyperfine coupling form factor as defined by Eq. (3) and Eq. (2) of SI. The point is that this form factor is **zero** at the K point (for the isotropic hyperfine coupling), and it is thus highly untrivial that below T_U the sum closely follows the behavior calculated for the K point (that is filtered out in the sum!), as shown in Fig. 2d.

3) The following sentences, employed **only** in the rebuttal letter, should definitely appear in the manuscript and/or SI: "Needless to say, the real material is always more complicated. For example, influences from higher CEF levels above the non-Kramers doublet, the interlayer couplings not included in our model calculations, and the lack of precise knowledge on the form factor, etc, may explain the subtle difference still remained between the panels (c) and (d) of the revised Fig. 2 in the manuscript."

4) In Fig. 2d, the calculated data obviously present significant errors-induced scatter, so that we **cannot** conclude that there exists a pronounced dip at T_U , contrary to what is suggested by the red solid line, supposed to be an **unbiased** guide to the eye.

5) I have a problem of understanding the Supplementary Fig. 2b: the relevant model is supposed to be a nucleus that is equidistant to the three spins on the vertices of an equilateral triangle and is coupled to them by isotropic hyperfine coupling, for which the corresponding $1/T_1$ form factor is proportional to: $1 + 4 \cos(x/2) [\cos(y \sqrt{3}/2) + \cos(x/2)]$. This appears to be what is shown **inside** the central hexagon of Supplementary Fig. 2b, and this pattern should be repeated everywhere in the x-y plane. That is, the (nearly) zero values given by the blue zones surrounding the central hexagon in the Supplementary Fig. 2b should not exist. Please check!

6) Minor details:

a) Page 2, column 2, 1st paragraph of "Results": "(Supplementary Fig. 2)" should be "(Supplementary Fig. 4)"

b) Page 3, column 1, 1st line: the old text "..., where the calculated $1/T_1$ exhibits an anomalous increase down to T_U , ..." should be updated as there are now **two** calculated curves in Fig.3d, so one should define which one is spoken about.

c) Page 3, column 2, line 5: "at the left side" should rather be "at lower fields"

- d) Page 3, column 2, line 9: "and below" should be deleted, because in Fig. 3 there are *no* data shown below 0.4 K.
- e) Page 3, column 2, line 13: "Moreover," should probably be deleted.
- f) Supplementary Information, page 4, line 6: "... whose first-order correction to the resonance frequency barely changes ..." is difficult to follow/understand even for a specialist. One should better be more explicit and say that the width of the line is supposed to be defined by the unresolved quadrupolar splitting of the line (into 3 lines for the nuclear spin 3/2), and therefore reflects the distribution of EFGs/quadrupolar couplings.
- g) Supplementary Information, page 5, note 5, 2nd paragraph: "The $1/69T_2$ for the out-of-plane fields are also shown in Supplementary Fig. 5, whose" appears to be repeating the previous sentence, and should probably be replaced by "These" (without the line break).

Reviewer #3:

Remarks to the Author:

The authors reply to my comments and concerns are satisfactory. They have added the missing data and explained the confusing points. As I mentioned before, NMR in a dilution refrigerator is not revolutionary but certainly not trivial and the data they obtained certainly deserved publication in nature communication. The comparison with theory is fair and justified, and in the new version the limitations of this comparison are more transparent. I therefore support publication of the manuscript in nature communication.

Response to the first Reviewer's report

Reviewer #1 : *In the revised version of the manuscript the authors took carefully account of the comments raised in my report and clarified a few open issues. I recommend the present version of the manuscript for publication in Nature Communications.*

Reply: We thank the reviewer for the recommendation of our work for publication.

Response to the second Reviewer's report

Reviewer #2 : *In the revised manuscript, the authors have taken into account the referees' remarks, and have, in particular, considerably improved the analysis by:*

A) adding the temperature dependence of the 2nd and the 3rd moment of the NMR spectra and by

B) taking into account the q-dependence of the hyperfine coupling tensor in the analysis of the $1/T_1$ data.

Notably, the later point B) improved the agreement of the theoretical prediction and measured NMR $1/T_1$ data (reflecting the low-energy spin fluctuations), so that the conclusions of the manuscript are now more convincing. However, regarding the modified parts of the manuscript I would have several comments and suggestion (given below) to improve the presentation. I would recommend the publication after the authors have taken them into account:

Reply: We thank Reviewer #2 for the positive evaluation and highly appreciate the valuable comments/suggestions below.

Reviewer #2 : *1) Regarding the information on the static/average spin distribution given by NMR spectra, from Fig. 2b and Supplementary Fig. 4, we learn that for the two putative BKT transitions the lower one at T_L is clearly visible *only* in the 1st moment, while the upper one at T_U is clearly visible *only* in the 3rd moment. This is a quite unexpected situation, which merits to be explicitly underlined/commented in the manuscript.*

Reply: We thank the Reviewer for this insightful comment. Indeed, the first moment, i.e. the hyperfine shift that reflects the magnetic susceptibility, exhibits a peak near the lower BKT transition, as the true long-range magnetic order develops below $T_L \sim 1$ K. On the other hand, both the 2nd (width of the NMR peak) and 3rd moments (asymmetry of the peak) actually change significantly near $T_U \sim 1.8$ K, which reflects the onset of a quasi-long-range order in the BKT phase. These experimental results evidence the two-step melting of magnetic order through the two BKT transitions, and it is the existence of the intermediate floating BKT phase in the system that gives rise to such unconventional behavior. Currently, we are not sure why the BKT transition at $T_L(T_U)$ is only obvious in the first(third) moment (and not the other way around), and leave it for future studies. Following Reviewer #2's advice, we describe these experimental observations in the revised manuscript.

Reviewer #2 : 2) The Fig. 2 caption should **explicitly** mention that the sum over q includes the hyperfine coupling form factor as defined by Eq. (3) and Eq. (2) of SI. The point is that this form factor is **zero** at the K point (for the isotropic hyperfine coupling), and it is thus highly untrivial that below T_U the sum closely follows the behavior calculated for the K point (that is filtered out in the sum!), as shown in Fig. 2d.

Reply: Thanks for the suggestion, and now we refer to the SI explicitly the hyperfine form in the caption of Fig. 2.

The grey line in Fig. 2d is not exactly at K but at its vicinity (K'). At low temperatures a diverging static peak arises right at the K point, and thus we switch to its vicinity (K') in Fig. 2d, where the low-energy excitations can be also well followed. Therefore, although K point itself is filtered by the form factor, the K' point around it is included (with nonzero weight in the form factor) and it does contribute to $1/T_1$ considerably because of the large low-energy spectral weight at K' point. Therefore, it is not a surprise that the two lines (sum and K') in Fig. 2d resemble each other below T_U .

Reviewer #2 : 3) The following sentences, employed **only** in the rebuttal letter, should definitely appear in the manuscript and/or SI: “Needless to say, the real material is always more complicated. For example, influences from higher CEF levels above the non-Kramers doublet, the interlayer couplings not included in our model calculations, and the lack of precise knowledge on the form factor, etc, may explain the subtle difference still remained between the panels (c) and (d) of the revised Fig. 2 in the manuscript.”

Reply: Thanks for this very good suggestion. Now they are included in the revised manuscript (in page 3).

Reviewer #2 : 4) In Fig. 2d, the calculated data obviously present significant errors-induced scatter, so that we **cannot** conclude that there exists a pronounced dip at T_U , contrary to what is suggested by the red solid line, supposed to be an **unbiased** guide to the eye.

Reply: Thanks for the comment. The red solid line in the previous version slightly exaggerated the dip near upper BKT transition. Now we have smoothed it (via higher-order polynomial fitting) to represent an averaged unbiased behavior of the scatters. As shown in the updated Fig. 2d, a (smooth) dip can still be clearly recognized.

Reviewer #2 : 5) I have a problem of understanding the Supplementary Fig. 2b: the relevant model is supposed to be a nucleus that is equidistant to the three spins on the vertices of an equilateral triangle and is coupled to them by isotropic hyperfine coupling, for which the corresponding $1/T_1$ form factor is proportional to: $1 + 4 \cos(\frac{x}{2})[\cos(\frac{\sqrt{3}y}{2}) + \cos(\frac{x}{2})]$. This appears to be what is shown **inside** the central hexagon of Supplementary Fig. 2b, and this pattern should be repeated everywhere in the $x - y$ plane. That is, the (nearly) zero values given by the blue zones surrounding the central hexagon in the Supplementary Fig. 2b should not exist. Please check!

Reply: Thanks for reminding. Yes, the form factor should be repeated in the hexagons around the

central one. The nearly zero values in the surrounding blue zones were due to improper extrapolation of the data to region outside the central hexagon, and indeed should be zero. We apologize for that and now have accordingly updated the Supplementary Fig. 2b.

Reviewer #2 : 6) *Minor details:*

a) Page 2, column 2, 1st paragraph of “Results”: “(Supplementary Fig. 2)” should be “(Supplementary Fig. 4)” .?

b) Page 3, column 1, 1st line: the old text “, where the calculated $1/T_1$ exhibits an anomalous increase down to T_U , ” should be updated as there are now *two* calculated curves in Fig.3d, so one should define which one is spoken about.

c) Page 3, column 2, line 5: “at the left side” should rather be “at lower fields”.

d) Page 3, column 2, line 9: “and below” should be deleted, because in Fig. 3 there are *no* data shown below 0.4 K.

e) Page 3, column 2, line 13: “Moreover,” should probably be deleted.

f) Supplementary Information, page 4, line 6: “ whose first-order correction to the resonance frequency barely changes ” is difficult to follow/understand even for a specialist. One should better be more explicit and say that the width of the line is supposed to be defined by the unresolved quadrupolar splitting of the line (into 3 lines for the nuclear spin 3/2), and therefore reflects the distribution of EFGs/quadrupolar couplings.

g) Supplementary Information, page 5, note 5, 2nd paragraph: “The 1/69 T_2 for the out-of-plane fields are also shown in Supplementary Fig. 5, whose” appears to be repeating the previous sentence, and should probably be replaced by “These” (without the line break).

Reply: We thank Reviewer #2 for the careful reviewing and for pointing out these typos and misprints. We have made revisions accordingly in the resubmitted manuscript.

Response to the third Reviewer’s report

Reviewer #3 : *The authors reply to my comments and concerns are satisfactory. They have added the missing data and explained the confusing points. As I mentioned before, NMR in a dilution refrigerator is not revolutionary but certainly not trivial and the data they obtained certainly deserved publication in nature communication. The comparison with theory is fair and justified, and in the new version the limitations of this comparison are more transparent. I therefore support publication of the manuscript in nature communication.*

Reply: We thank the Reviewer for his/her positive assessment and the recommendation of our revised manuscript.

Summary of changes

1. Main text, Page 2, right column, 1st paragraph: we add a sentence discussing the different behaviors of the first, second, and third moment of NMR spectra, and their connections to the BKT phase.
2. Main text, Page 3, left column, last paragraph: we add some discussions on the complication of real material and point out possible reasons that can explain the subtle difference between the model calculation and experimental measurements.
3. Previous errors associated with the data points in Fig. 2b were not determined properly. We have now corrected it and updated Fig. 2b.
4. Fig. 2d of the main text is updated, with a smoothed solid line as guide to the eyes, and the K' point where $1/T_1$ was computed also labeled explicitly.
5. Supplementary Fig. 2b is replotted, and Supplementary Note 2 is revised accordingly.
6. In the Supplementary Note 4, the first paragraph is revised following the suggestions from Reviewer #2, with the the high-temperature NMR linewidth as the distribution of the EFG couplings spoken out explicitly.
7. Some other typos in the main text and the SI are corrected and marked by blue color.